ecology/environmental science

conservation translocation, species distribution models, assisted migration, ecological modelling, Australian mega-fires

**Author for correspondence:**
Shane D. Morris
e-mail: shane.morris@utas.edu.au

# Roughing it: terrain is crucial in identifying novel translocation sites for the vulnerable brush-tailed rock-wallaby (*Petrogale pencillata*)

Shane D. Morris[1], Christopher N. Johnson[1,2] and Barry W. Brook[1,2]

[1]School of Natural Sciences, University of Tasmania, Private Bag 55, Hobart, Tasmania 7001, Australia
[2]ARC Centre of Excellence for Australian Biodiversity and Heritage, University of Tasmania, Hobart, Tasmania 7001, Australia

SDM, 0000-0002-7026-5878; CNJ, 0000-0002-9719-3771; BWB, 0000-0002-2491-1517

Translocations—the movement of species from one place to another—are likely to become more common as conservation attempts to protect small isolated populations from threats posed by extreme events such as bushfires. The recent Australian mega-fires burnt almost 40% of the habitat of the brush-tailed rock-wallaby (*Petrogale pencillata*), a threatened species whose distribution is already restricted, primarily due to predation by invasive species. This chronic threat of over-predation, coupled with the possible extinction of the genetically distinct southern population (approx. 40 individuals in the wild), makes this species a candidate for a conservation translocation. Here, we use species distribution models to identify translocation sites for the brush-tailed rock-wallaby. Our models exhibited high predictive accuracy, and show that terrain roughness, a surrogate for predator refugia, is the most important variable. Tasmania, which currently has no rock-wallabies, showed high suitability and is fox-free, making it a promising candidate site. We outline our argument for the trial translocation of rock-wallaby to Maria Island, located off Tasmania's eastern coast. This research offers a transparent assessment of the translocation potential of a threatened species, which can be adapted to other taxa and systems.

# 1. Introduction

The vulnerability of small populations to stochastic events has been thoroughly documented in the ecological literature [1–4]. Large stochastic events with potentially harmful consequences (bushfires, floods, heat waves etc.) are expected to increase with climate change [5]. Conservation translocation—the intentional movement of species from one place to another for conservation purposes—is likely to become more common in response to the increasing threat posed by such extreme events [6]. The identification of potential translocation sites is a key first step in this process. In the past, these have been chosen by locating areas with similar habitat characteristics to those currently or recently occupied by the species [7,8], the rationale being that species will have a greater chance of survival and persistence in areas that are bioclimatically and environmentally similar to their existing range. Correlative species distribution models (SDMs) can accomplish this task cheaply, quickly, and with quantitative confidence.

Correlative SDMs are developed by fitting statistical relationships between presence records and selected predictors (climatic, geological, biological etc.) [9] to identify areas in environment-geographical space with high similarity to the species' occurrence data. They have been used to identify potential sites for a myriad of species, from mussels (*Margaritifera margaritifera*) [10] to European bison (*Bison bonasus*) [11]. In plants, areas which were predicted by SDMs to have higher suitability resulted in higher germination success post-translocation [12].

Notwithstanding these benefits, the reliability of SDMs is often compromised by their assumption that species are in equilibrium with their habitat, and the exclusion of factors such as dispersal limitations (though this has been incorporated in newer models, see Zurell *et al.* [13] for further details), and biotic interactions that may vary spatially and temporally. As a result, SDMs are not always suitable for extrapolation into new conditions [14]. These problems are particularly acute when predicting future distributions, which necessitates both spatial and temporal extrapolation. However, when constraining the projections to current climate conditions but geographically different areas, correlative SDMs can perform as well as process-explicit or mechanistic models [13,15,16]. This makes them a useful 'first filter' for identifying potential translocation sites for threatened species needing immediate translocation [17,18]. One such species is the brush-tailed rock-wallaby (*Petrogale penicillata*, hereafter rock-wallaby), a medium-sized (5–11 kg) marsupial currently classified as Vulnerable [19]. Initial estimates state that 38% of the rock-wallaby's habitat was burnt during the recent 2019/2020 Australian wildfires, though the effect of this on their metapopulation is yet to be revealed [20]. This highlights the need for the establishment of additional 'insurance populations', to lessen the risk of further population decline; translocation may be a viable way to achieve this.

Before European colonization of Australia, the rock-wallaby ranged in rocky terrain across the forests and woodlands in the southeast (SE) of the country [21] from southeastern Queensland, through New South Wales (NSW), and into Victoria [22,23]. The current distribution reflects a combination of past and present threats (figure 1). Invasive predators, primarily the European red fox (*Vulpes vulpes*), are the current major threat and have restricted rock-wallabies to inaccessible rocky refuges along the great dividing range (GDR) [22–24,26,27]. Additionally, foxes have limited the connectedness of colonies because wallabies are susceptible to predation in open areas during dispersal. This has increased inbreeding effects [27]. Also, habitat loss, habitat degradation and hunting led to the contraction of their range as vegetation was cleared, domesticated herbivores introduced, and animals were killed *en masse* for their meat and fur [22,23,26,27]. It is estimated that half a million animals were killed in NSW during 30 years at the beginning of the twentieth century [28].

The brush-tailed rock-wallaby metapopulation exhibits natural segregation, being split into three evolutionary significant units (ESUs)—the Northern (north of approx. 32° S in figure 1), Central (between 32° S and 35° S) and Southern (south of 35° S)—which are genetically distinct from one another [24,27]. The Southern ESU is the most threatened. The westernmost population in this ESU, in the Grampian mountains (figure 1), went extinct in 1999 and was re-established by a translocation, composed of members of the Southern ESU and New Zealand populations (see below), between 2008 and 2012 [29]. This reintroduction consisted of 39 individuals, one-third of which were killed by predators, probably foxes, soon after release [29]. Previous translocations, undertaken for non-conservation purposes, established populations in New Zealand and Hawaii. The former was deliberately released in 1863 by the then Governor of New Zealand [30], the latter escaped from captivity in 1916 [31].

Previously, SDMs have been constructed for the rock-wallaby at the regional scale. Murray *et al.* [32] assessed the transferability of site- and landscape-level predictors, and found that habitat complexity (the number of refuges, ledges etc.) was the best predictor of habitat and that the landscape-level predictors (slope, geology, land cover, remnant vegetation) had, by comparison, poor transferability. One way to

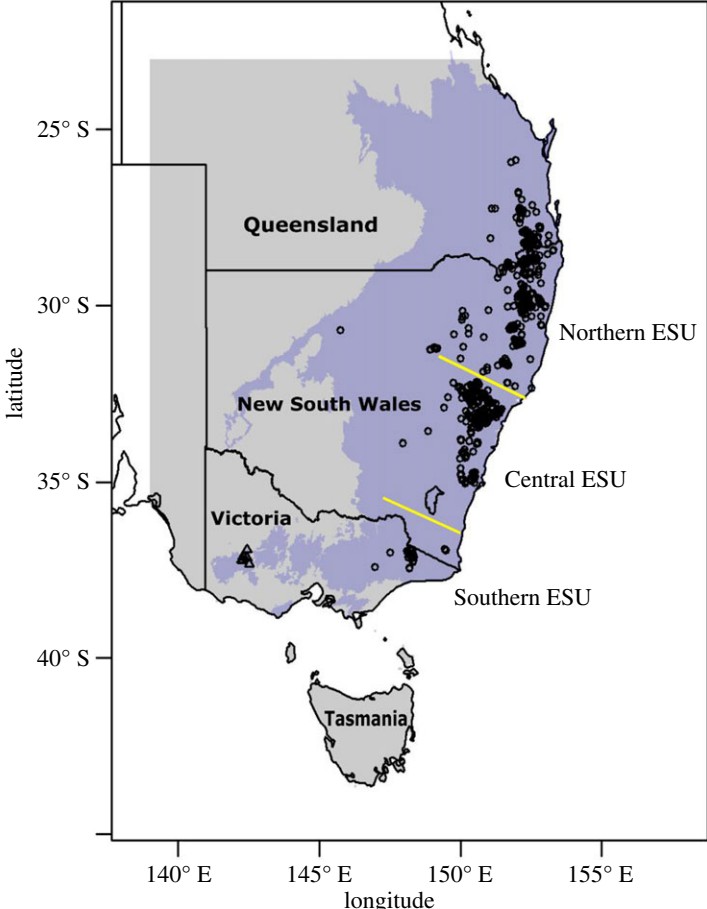

**Figure 1.** A map of occurrences of the brush-tailed rock-wallaby across southeast Australia. The black circles ($n = 1400$) indicate the occurrence points used to create the Maxent-based species distribution model, while the black triangles ($n = 9$) are the records in the Grampians used for validation. The purple area is the biogeographic regions that make up the species' natural range [24] plus the Australian Alps region as it has recorded occurrences (see Methods). The grey area is the southeast background. The location of the separation between the evolutionary significant units (ESUs) is based on Eldridge *et al.* [25].

approximate habitat complexity via remote sensing is to use terrain roughness, along with continuous measures of land use and vegetation (woodland and grassland). Ashcroft *et al.* [33] showed that composite topoclimatic predictors should be favoured, when available, over individual macroclimatic and topographic predictors, because they better capture microclimatic phenomena such as cold-air drainage. However, the calculation and validation of topoclimatic data are not currently feasible at a subcontinental scale. An alternative is to use the percentage of north-facing sites, minimum temperature in the coldest month, and annual mean precipitation to approximate the effect of climate on rock-wallaby distribution. Two climatic variables were chosen to reduce collinearity and the risk of overfitting. The two chosen—minimum temperature in the coldest month and annual mean precipitation—were selected as rock-wallabies have previously been shown to be negatively affected by cold conditions and high precipitation [28,33].

Here, we use maximum-entropy SDMs to identify new potential translocation sites for the rock-wallaby across southeastern Australia, including sites within the large island of Tasmania, beyond their historical range. Models were internally evaluated using cross-validation, weighted on their performance on out-of-sampling data, ensembled and projected across the SE of Australia, with the goal of assessing the areas with the highest habitat suitability that might be suitable for future conservation translocations.

## 2. Methods

We extracted brush-tailed rock-wallaby presence data from the Atlas of Living Australia (ALA: www.ala.org.au) on 17 April 2020. Only presence records from Australia were used (i.e. excluding the records of

introduced wallabies in New Zealand). Data provided by 'Western Australian Museum provider for OZCAM' were also excluded as they had four erroneous occurrences that did not agree with the accepted range of the species [24]. To avoid the inclusion of morphologically similar species (yellow-footed rock-wallaby (*Petrogale xanthopus*), allied rock-wallaby (*P. assimilis*), Herbert's rock-wallaby (*P. herberti*) etc.), crowdsourced data (OzAtlas, iNaturalist etc.) were excluded. The total number of occurrences extracted was 4576. Only occurrences with an accuracy of 1 km or less were retained, as a single entry per each latitude and longitude pairing ($n = 1415$). These were visually inspected for any likely mistakes and eight were removed. Five were excluded as they occurred in urban environs (e.g. Sydney, Wagga Wagga), two were from the nineteenth century in Western Australia and are probably a different species, perhaps the black-flanked rock-wallaby (*P. lateralis lateralis*), and a final entry from Queensland as it was probably Herbert's rock-wallaby. To validate the predictions of our model, the Grampians records (nine occurrences) were excluded, the rationale being that if the final model predicted high suitability here, we could have more confidence in it. The final total of occurrences fitted in the modelling was 1400. Projections were done across the SE of Australia, including Tasmania, while other regions (SE, northwest and northeast of the continent) were excluded because they have other rock-wallaby species, ruling them out as potential translocation sites due to the potential for competition with resident species.

Maxent models were used for fitting the SDMs, because they are designed specifically for presence-only data [34], have a proven predictive performance [35], and are recommended when the goal of the study is not model transfer [36,37]. Maxent uses background points, in lieu of absence points, to estimate the prevalence of conditions (range of predictors) to which it can compare the occurrence points. The choice of background points can affect the predictions of the model [37], as they change the relationship of the occurrence to the prevalence of the conditions. To illustrate: if the mean temperature of a species' distribution is 8°C, background points with a mean temperature of 6°C will predict this species to inhabit warmer areas, whereas background points with a mean of 15°C will predict it as a cold-dwelling species. Restricting background points to areas reachable to the species, i.e. those devoid of dispersal barriers, is the most ecological realistic choice [36], but is constrained in its ability to extrapolate outside those conditions. Therefore, we generated background points from two spatial extents, 30 000 points from across the SE of Australia (including Tasmania, referred to as SE) and 10 000 background points in all the Interim Biogeographic Regionalisation for Australia (IBRA, Department of the Environment [38]) regions which are deemed to be part of the species' natural range [24] (similar to Briscoe *et al.* [16]). The Australian Alps bioregion was included in the latter extent as occurrences from the ALA were located there. We chose 10 000 background points based on recommendations from Barbet-Massin *et al.* [39] and chose 30 000 to control for the larger extent of the SE background. An Albers projection of Australia was used, as it is an equal-area projection and Maxent assumes equal cell size [36].

Six ecologically relevant predictors—terrain roughness (a surrogate for rockiness and availability of predator refugia), aspect, percentage cover of woodland, percentage cover of grassland (and other non-woody vegetation), minimum temperature in the coldest month (July) and mean annual precipitation—were selected to identify potential translocation sites. This is a lower number of variables than typically used for SDMs, but the use of too many variables risks overfitting, collinearity and loss of relevance to the species [40]. Terrain roughness, which is the difference between the maximum and minimum elevation of a raster cell and its surrounding eight cells, was calculated using the `terrain` function in the `raster` package [41] on the GEODATA 9 Second Digital Elevation Model (DEM-9S) v. 3, which covers Australia at a 250 m resolution. It was selected as rock-wallabies need complex terrain for shelter and to hide from predators [26]. Previous studies [26,32,33] used elevation and slope to model inaccessible terrain. Terrain roughness acts as a composite of the two as areas with high values will probably have cells of high elevation with areas of much lower elevation nearby meaning steeper inclines. Aspect, the direction in which a landscape is oriented, was found to be an important determinant for favourable habitat in NSW [22], with rock-wallabies more likely to be found on north-facing slopes due to their warmth. Although Murray *et al.* [26] warn against transferring site variables to larger-scale studies, the underlying mechanism (the seeking of warmer sites) is likely to remain relevant at larger scales. Aspect was extracted again using `terrain` function and then converted to a percentage based on the number of north-facing 250 m cells relative to the larger approximately 1 km cell.

The national vegetation information system (NVIS) v. 5.1 for Major Vegetation Groups [42], which covers Australia at a 100 m resolution, was used to derive the woodland and grassland data. Both were deemed necessary because rock-wallabies forage in forests, woodlands and grasslands [27]. We created two spatial layers, one representing the percentage of woodland/forest in a grid cell (0.01°, or

approx. 1 km) and the other the percentage of grassland, shrublands, sedgelands (etc.) (for collapsing of NVIS categories see electronic supplementary material, S1). All mangroves, water bodies, cleared non-native vegetation, buildings and unknown data were omitted from these calculations. Cleared non-native vegetation was used as a variable in previous studies [26,32]; however, the resolution of our vegetation data was not high enough to distinguish between human infrastructure and cleared non-native vegetation, so we chose to remove both. The final predictors were minimum temperature of the coldest month and mean annual precipitation because it is well known that climate can dictate species distribution at large spatial scales [43–45]. These were extracted from the ANUClim dataset [46], which contains averaged macroclimatic data for Australia, across a 30-year period (1976–2005), at an approximately 1 km resolution. Rock-wallabies are believed to be susceptible to cold conditions and excess precipitation, given that they are less active on wet nights in cold regions [28] and have contracted most from the colder part of their range (increased basking reduces time for foraging and increases susceptibility to predation) [33]. These variables were also chosen to avoid overprediction in colder and wetter climes (e.g. Tasmania). The correlation between these variables was inspected using Spearman's rank correlation, as collinearity is a common problem in SDM construction [47], although Maxent is less influenced by this than other algorithms [37].

Maxent models were parametrized with only hinge features allowed, and a beta-value of 2.5, allowing a partly nonlinear fit [36,48] that would be best suited to generalizing to new areas. Hinge features can model complex relationships (e.g. inverted U-shape) and have strong predictive accuracy [49]. As rock-wallabies are difficult to detect [50], prevalence was set as 0.3. Models were k-fold cross-validated ($k = 10$) and this k-fold-sampling was repeated 10 times to ensure a robust and stable model-selection metric. All of the best models derived from each k-fold sampling were then ensembled and weighted using area under the receiver operating curve (AUC). AUC, in a Maxent context, is a measure of a model's ability to discriminate between the presence and the background points [37]. True skill statistic (TSS), which is a measure of accuracy developed specifically for SDMs, was also calculated [51]. All models were fitted, evaluated and ensembled using the `sdm` package [52]. A single Maxent model, with the same settings as above, was also created and tested using the `limiting` function in the `Rmaxent` package [53], to visualize the limiting predictors of rock-wallabies spatially.

# 3. Results

The six predictors showed minimal intercorrelation, with the greatest being between the minimum temperature in July and the mean annual precipitation (Spearman rank coefficient, SRC = 0.53). The largest negative correlation was between terrain roughness and minimum temperature in July (SRC = −0.44). Terrain roughness was the most important predictor—by a substantial margin—in correctly identifying suitable areas (figure 2a), while aspect was the least. The method of background selection did not affect the importance ranking of the predictors, but it did affect the response curves (figure 2b). The response curves of percentage woodland and the minimum temperature in July changed the most, with both curves flattening when the full SE region was evaluated. Percentage woodland had a positive effect when the IBRA background was used, but that effect was not discernible when the whole of the SE was used as background. Minimum temperature in July displayed a unimodal response with a mid-domain peak at 5°C when the IBRA regions were used as the background. The response curve was flat when the larger background area was used. Terrain roughness had a positive effect up until values of 220 m, after which its response dropped in strength. The response curve of mean annual precipitation was again unimodal, with values of below 500 mm and above 1100 mm associated with low response values. As aspect and percentage of grassland had low importance, their response curves were flat, as expected.

The ensembled Maxent model for the IBRA background and SE background had AUC scores of 0.96 and 0.98, respectively, and TSS of 0.81 and 0.89. However, as we omitted the occurrences in the Grampians, these could act as an external validation of our models' performances. The median relative habitat suitability scores predicted for these hold-out points were 0.56 (range 0.01–0.65, IBRA background) and 0.51 (range 0.01–0.62, SE background). By contrast, the suitability scores of 10 000 groups of nine spatial points were extracted for each background to assess the probability of getting these Grampians values by chance. The mean suitability score for these 10 000 'null' groups was 0.0192 (95% confidence interval (CI): 0.0185–0.0199, IBRA) and 0.0014 (95% CI: 0.0012–0.0015, SE). That is, the probability of obtaining our suitability estimates for the Grampians, by chance, was indistinguishable from zero for both backgrounds.

(a)

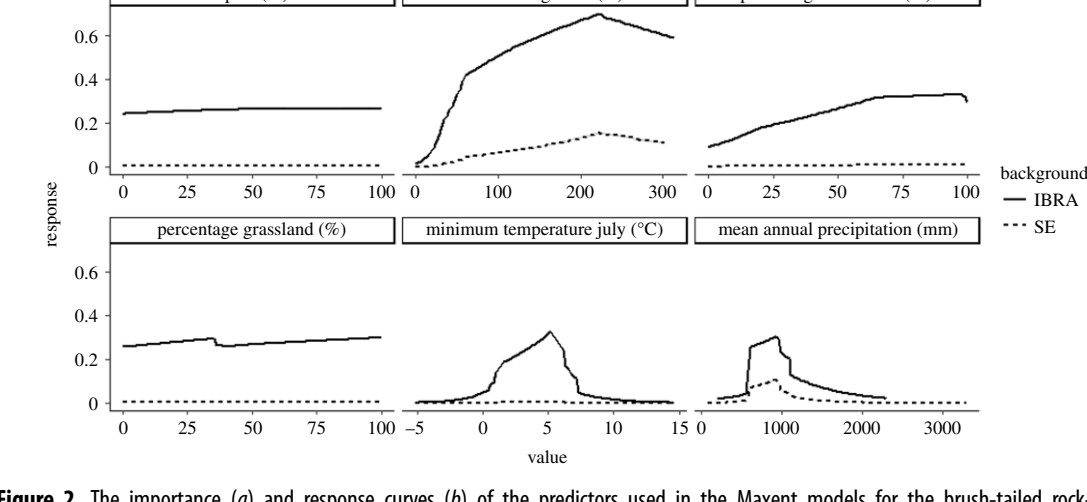

(b)

**Figure 2.** The importance (a) and response curves (b) of the predictors used in the Maxent models for the brush-tailed rock-wallaby. The dark grey columns (a) and solid lines (b) are the results for the predictors when the IBRA bioregions were used to select the pseudo-absences, while the light grey columns and dashed line are the results when the entire southeastern portion of Australia was used (SE).

The model did not predict many areas of high suitability far from areas the rock-wallaby currently occupies (figure 3a). The exceptions to this were the Yarra Ranges (latitude approx. 37.5° S, longitude approx. 146° E), the Central Queensland Sandstone Belt (latitude approx. 25° S, longitude approx. 148° E) and Tasmania. In fact, a large portion of eastern Tasmania was considered suitable. The choice of background affected the magnitude of the suitability, the IBRA background predicted higher suitabilities than the SE background, but overall, the two alternative background-sampling approaches showed no major discrepancies (figure 3b).

Figure 3c shows the spatial distribution of the limiting predictors for rock-wallaby suitability. Mean annual precipitation contributed most to the classification of the interior as unsuitable (it is too dry) while terrain ruggedness was a large contributor from the sub-interior towards the coast (including most of the Grampians). Along the GDR, minimum temperature in July was the limiting factor in the model. Precipitation again was the limiting factor in western Tasmania, although this time because of high rainfall.

## 4. Discussion

Our projections of the distribution of the brush-tailed rock-wallaby across the SE of Australia illustrate the limited availability of unoccupied areas with high suitability, implying that relatively few potential translocation sites exist. Mainland areas with high suitability which do not currently contain brush-tailed rock-wallabies—the Yarra Ranges and the Central Queensland Sandstone Belt—might be risky

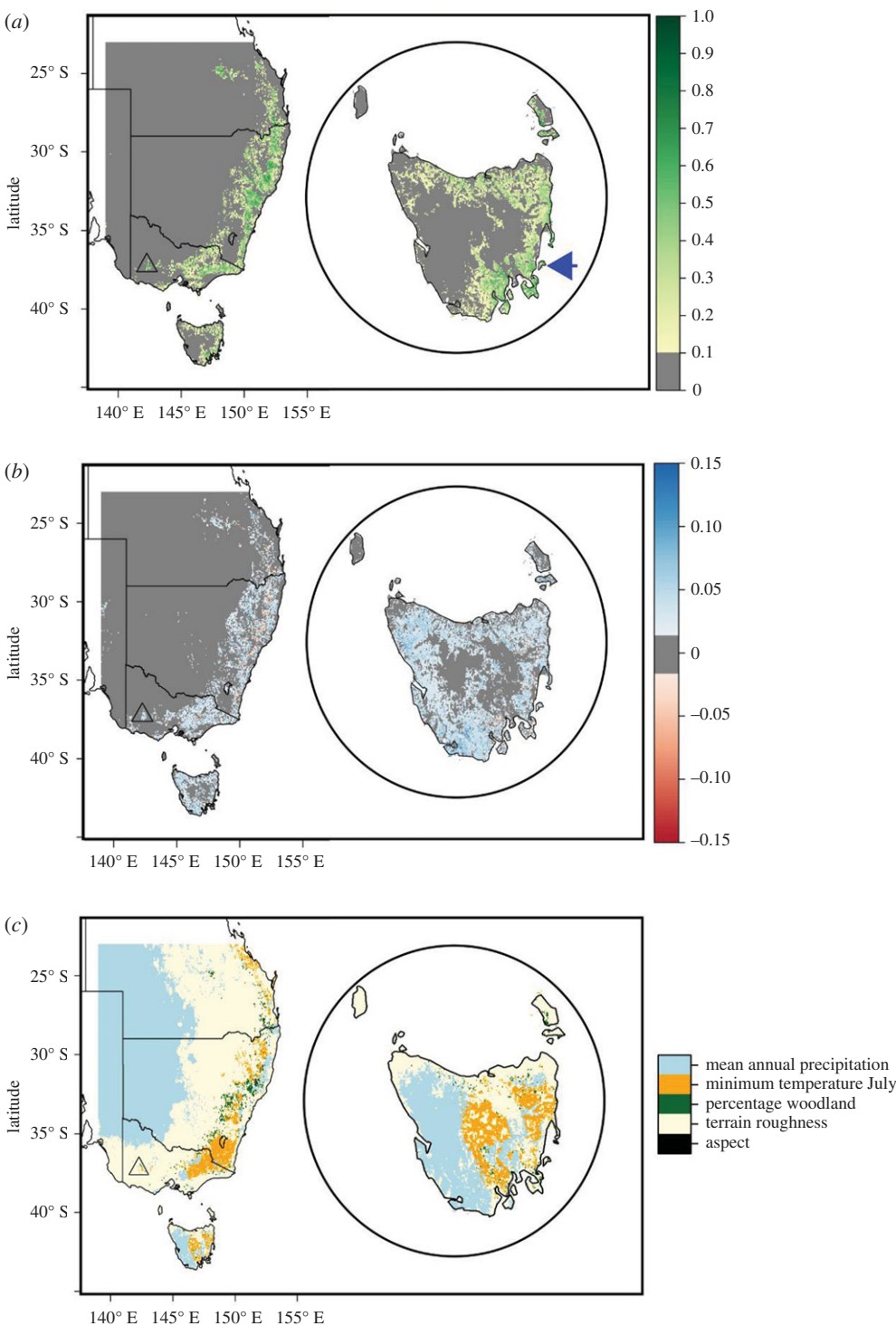

**Figure 3.** The spatial distribution of habitat suitability (*a*), the change in suitability using different background points (*b*), and factors limiting occurrence (*c*), for the brush-tailed rock-wallaby across southeastern Australia and Tasmania (inset). The location of the Grampians is represented by the black triangle in each map while Maria island is indicated by the arrow in the inset map in (*a*). (*a*) was created using background points for the Maxent model selected from the biogeographic regions where rock-wallabies occur. (*b*) shows the difference between the habitat suitability in (*a*) and that when the whole study was used for background point selection, with blue indicating areas where (*a*) predicted higher suitability and red where it predicted lower.

places for translocations, given that the former supports a large population of foxes [54] and the latter contains Herbert's rock-wallaby [55]. The role of foxes in the failure of the Grampians translocation [29] highlights the need for release sites at which foxes are absent, or are under high levels of control/

suppression. The eastern half of Tasmania displayed high suitability and is almost certainly fox-free. Therefore, Tasmania would make a promising candidate for a recipient site.

A major drawback for Tasmania as a potential translocation site is the absence of any ecologically similar species [28]. There is, as yet, no fossil evidence of any *Petrogale* spp. inhabiting Tasmania during the Late Quaternary [56]. Moving a species, due to conservation concerns, into an area it has never occupied before is termed an 'assisted migration' (or assisted colonization [6]), and is viewed as the riskiest of translocation procedures, due primarily to the unknown ecological consequences that might be triggered [57]. Rock-wallabies are mixed feeders that consume grasses, forbs and shrubs [58], so they could potentially exhibit novel browsing pressure, damaging native vegetation. No comparison of diet can be made between the brush-tailed rock-wallaby and Tasmania's two similar-sized macropods, the Tasmanian pademelon (*Thylogale billardierii*) and Bennett's wallaby (*Macropus rufogriseus rufogriseus*), because no detailed dietary assessment exists for these two species in Tasmania, although it does for the rock-wallaby in NSW [59]. There is limited evidence, from northern NSW, which suggests little dietary overlap (11.9%) between the red-necked wallaby (*Macropus rufogriseus banksianus*) and brush-tailed rock-wallaby [58], though this should be tested for the Tasmanian subspecies of *M. rufogriseus*.

Despite the risks, the rock-wallaby meets the three criteria required for an assisted migration to be considered: the need to move, the potential to move, and the inability to move in the face of global change [60]. The incessant reduction of rock-wallaby populations over the last century, compounded by the recent (2019/2020) bushfires, have created a need. The results of this study support the potential of Tasmania as a recipient site, and the need for direct conservation intervention is clear, due to the oceanic dispersal barrier (Bass Strait) that prevents any natural movement from the mainland to Tasmania.

Past translocations of the brush-tailed rock-wallaby, to New Zealand and Hawaii [27], can shed light on the potential consequences of moving rock-wallabies to a novel, fox-free environment. In New Zealand (an island group with no native non-volant mammals), rock-wallabies are considered pests because they over-browse native vegetation (particularly *Metrosideros* spp.). As a result, populations are being eradicated, with two of the three populations (Motutapu and Rangitoto Islands) complete and the final one (Kawau Island) underway [27,61]. In Hawaii, the population has remained restricted to one valley (7 ha) [31]. Anecdotal evidence suggests that a predator pit created by feral dogs is the major reason for this restricted distribution [62]. If so, the containment of an introduced population of rock-wallabies to a recipient site might be reliant on predators either directly killing individuals, suppressing population growth, and/or inhibiting dispersal.

The Tasmanian devil (*Sarcophilus harrisii*), the largest extant carnivorous marsupial [63], could act in this role, if rock-wallabies were introduced to Tasmania, by reducing the probability of dispersal beyond rocky refuge areas to which they are introduced. Even if adult rock-wallabies could evade devils, the population would probably be limited to rocky areas. Juvenile rock-wallabies are left to shelter within rocky refuges while their mother feeds, leaving them vulnerable to predators [64,65]. If populations extend beyond these refuges their vulnerability increases because of lower security of available refuges, leading to higher levels of mortality. If a translocation to Tasmania was initiated, this should be to sites in the eastern part of Tasmania, which our modelling showed had the highest levels of suitability, while extreme values of temperature and rainfall limited suitability (figure 2*b*) in the west (figure 3*c*), supporting previous findings [33]. Although not considered here, climate change could increase the suitability of Tasmania even further as the island is expected to experience warmer temperatures, a 2.9°C increase in mean annual temperature by 2100 under a high emissions scenario, and unchanged mean annual precipitation state-wide, but with differences at the regional and seasonal level [66].

To allay concerns of unwanted negative consequences of a translocation, a trial translocation to an offshore island, such as Maria Island off eastern Tasmania (figure 3*a*, inset), might be the first potential step. Maria Island is a 9650 ha island, entirely a National Park, and has sections of steep rocky terrain, with two mountains over 600 m [67,68]. It has much potential, being larger and having areas of greater elevation than Motutapu and Rangitoto Islands, which sustained populations of at least 3500 and 8500, respectively [61]. See figure 4 for the concentration of suitable areas in areas of high elevation on Maria Island. However, populations would be unlikely to achieve these numbers as the island also has a population of Tasmanian devils that was introduced in 2012 as an insurance population after large declines on the Tasmanian mainland due to devil facial tumour disease [67]. Such a translocation would provide abundant opportunities for research, e.g. the persistence of 'wariness' of predators from a species' evolutionary past, as the last time these two species would have interacted would have been in the mid-Holocene before the devil's mainland extinction [70]. If approved by stakeholders, the translocation of the rock-wallaby could happen relatively quickly as

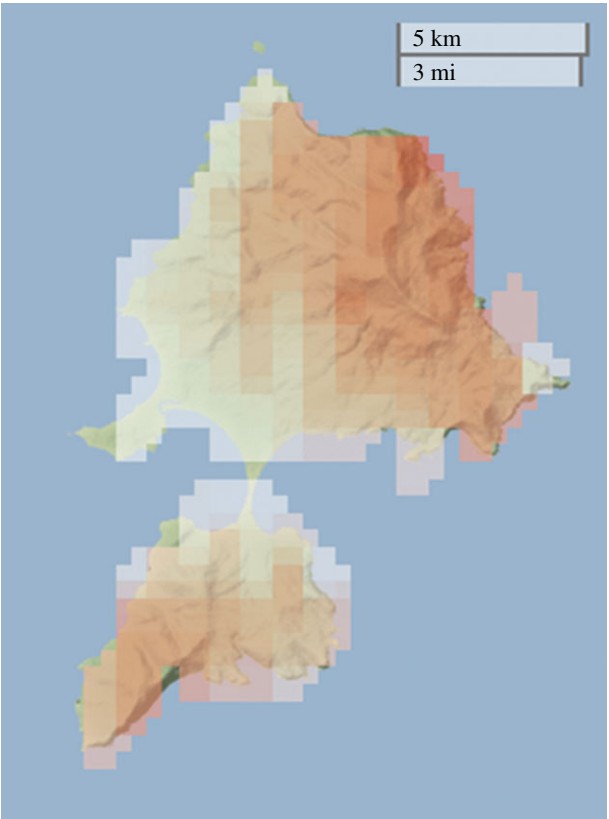

**Figure 4.** Map of the suitable areas of release for the brush-tailed rock-wallaby on Maria Island. High suitability is indicated by red and low suitability by white. Note the association of high suitability with the inaccessible terrain on the island. Produced using the R package leaflet [69].

there is already a captive breeding programme in place [27]. The matter is urgent, especially if the goal is to prevent the extinction of the Southern ESU because the latest population estimate was 80 individuals, only half of which were in the wild [29].

In addition to identifying Tasmania as a potential translocation region, our study assessed the effect of our chosen predictors on the present-day distribution of the rock-wallaby. Unlike previous studies [26,32,33], which used altitude and slope as predictors, we used terrain roughness. As expected, this proved to be by far the most important predictor in determining rock-wallaby habitat suitability (figure 2a), as it probably captures the characteristics of inaccessible habitat better than either elevation or slope alone. The greater importance of mean annual precipitation in comparison with minimum temperature supports previous findings [33] and suggests the importance of an availability of refuges in which to shelter, if a translocation to Tasmania were to take place. The importance of terrain roughness, which is probably a proxy for protection from predation, and rainfall (figure 2a), have been previously noted as the most influential factors dictating macropod populations [50,71–73].

The percentage of woodland cover was more important than the percentage of grassland cover, indicating the beneficial role that habitat complexity may play in rock-wallaby occurrence, though this might be affected by our exclusion of cleared land from the grassland calculations. Aspect was of little importance in predicting habitat suitability at this scale, but seems to be of greater importance at finer resolutions [22,26]. The relevance of scale in the choice of predictors for rock-wallaby habitat has been highlighted in detail before [26], and confidence in our choice of predictors is supported by the suitability projected for the Grampians. Figure 3c shows that terrain roughness is the limiting factor here, indicating the accessibility of this site to foxes as the possible cause of extinction and the failed reintroduction. However, caution is needed as our occurrences are heavily biased towards modern data. Rock-wallaby requirements of habitat complexity are evidently less strict in fox-free areas (for example, in New Zealand). Our projections are probably reliant on terrain roughness as a predictor for this reason. The use of a restricted population as the input data for Maxent can lead to the incorrect attribution of a species' habitat use [74]. The response curve for terrain roughness indicates a drop in suitability at high values but this may be due to these areas being less accessible to people

rather than to rock-wallabies. Nonetheless, terrain roughness should be considered as a predictor in future SDMs of other rock-wallaby species, as many are threatened by foxes or feral cats (*Felis catus*) [24]. If translocation is considered for these species, then fine-scale habitat features (as in Short [22]) will need to be identified, especially in areas where foxes are still present.

Ultimately, the undertaking of an assisted migration for the brush-tailed rock-wallaby does not depend solely on the predicted suitability of an area. It must not only be beneficial for the species, present a limited threat of invasive potential, and be logistically feasible; it must also be accepted and wanted by society [75]. Therefore, the future of the rock-wallaby will not depend on the AUC of an SDM, but also in large part on the decision making of the Australian people in the context of the trade-offs of twenty-first century conservation management.

.

Data accessibility. The data and code used to run this experiment are available at https://doi.org/10.5281/zenodo.4017016.

Authors' contributions. S.D.M., C.N.J. and B.W.B. conceived of the original idea; S.D.M. carried out the data collection and coding with critical input from C.N.J. and B.W.B. All authors contributed to the writing of this publication and gave final approval for publication.

Competing interests. The authors declare no competing interests.

Funding. This work was funded by Australian Research Council grant no. FL160100101 to B.W.B.

Acknowledgements. The authors are grateful to all the people who collected and collated the data used in this research, especially the occurrence data. We acknowledge the traditional owners of the land on which this data was collection.

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
