## [Reviewer comments · Royal Society Open Science]

Review History

RSOS-201603.R0 (Original submission)

Review form: Reviewer 1

Is the manuscript scientifically sound in its present form?

Yes

Are the interpretations and conclusions justified by the results?

Yes

Is the language acceptable?

Yes

Do you have any ethical concerns with this paper?

No

Have you any concerns about statistical analyses in this paper?

No

Recommendation?

Accept with minor revision (please list in comments)

Comments to the Author(s)

This is a nicely constructed paper which provides a good model for others wishing to use species distribution models for conservation planning. While the conclusions in this case are unsurprising, they do further refine understanding of the habitat of the Brush-tailed rock-wallaby, and the discussion on potential for introduction to areas outside the current distribution to provide protection from affects of climate change is topical.

I have only a few minor editorial comments and suggestions:

Line 67 – should be wallaby's, not wallabies.

Lines 107-108 – more information is needed on why these two climatic variables [minimum temperature in the coldest month, and annual mean precipitation] were selected from the large pool of potential climate variables.

Line 133 – Sup xxx – to be completed

Line 232 – apostrophe misplaced.

Line 233 – 'score' should be plural – there are 2, one for each background.

Line 245 – delete 'with'

Line 279 – sentence needs recasting as it currently suggests that a dietary assessment from Tasmania exists for the rock-wallaby. The dietary assessment referred to took place at three sites in NSW.

Line 287-288 – need to more directly link your expression 'the habitat and climate argument' to the second criterion for assisted migration – the potential to move – assuming that is what is meant.

Line 304 – delete 'outside their recipient site', it just repeats the information contributed by 'areas to which they were introduced'.

Line 314 – insert 'in' after 'increase'.

Line 320 – sentence beginning with 'It' needs work – 'offers', larger area of potential ?

Line 351 – delete 'that' or change 'as' to 'is'.

Line 352 – the terrain in the Grampians is extremely rough and broken. I don't see that rock-wallabies would be more exposed to predators there than at other occupied sites.

Line 354 – recast to 'being less accessible to people.'

References - need a careful check, for example the surname Burbidge is missing from number 19

Review form: Reviewer 2

Is the manuscript scientifically sound in its present form?

Yes

Are the interpretations and conclusions justified by the results?

Yes

Is the language acceptable?

Yes

Do you have any ethical concerns with this paper?

No

Have you any concerns about statistical analyses in this paper?

No

Recommendation?

Accept with minor revision (please list in comments)

Comments to the Author(s)

The authors describe a study where they used species distribution modeling to define areas to which the brush-tailed rock wallaby could be introduced since its original habitat is being diminished as a consequence of climate change effects.

In general, the manuscript is very well and clearly written, the methods are well justified and described adequately, and the results and conclusions inferred are robust. I see no major concern with the scientific content of the paper.

The study is case-specific and concerns the immediate conservation of a specific species for a specific reason. As such, I feel that the authors could have the potential to reach a more targeted audience through offering the manuscript for publication to a more conservation-oriented paper, like *Conservation Biology* where it could function as an interesting case-example for similar conservation projects. However, with a bit of effort and considering the open access policy of RSOS, I am sure that the paper has potential to reach readers most interested in this type of case study. The most novel aspect of their paper is the use of terrain characteristics in addition to habitat and climate variables, for modelling the suitable area for the species. As such, it offers a valuable example of how the scientific process can be used for informing practical conservation planning.

I have only a few minor comments, most of which relate to being a bit more specific with the concepts under the rather wide conservation translocation umbrella.

Minor comments:

line 24: could you be more specific on what kind of translocation you refer to? Translocation is an umbrella concept and includes very many different methods. Please refer to the IUCN: <https://www.iucn.org/content/guidelines-reintroductions-and-other-conservation-translocations> A translocation can be unintentional (cf. invasive species) or conducted because of conservation concern (=conservation translocation). I believe you are referring to a conservation introduction, and within that, assisted migration. Throughout the manuscript, please consider where you need to specify these terms more precisely.

line 40. Ditto. This definition is not specific enough, please refer to the IUCN guidelines: "Conservation translocation is the intentional movement and release of a living organism where the primary objective is a conservation benefit."

line 210: visual  visualize?

271-274: Again, please be specific with the definitions. Assisted migration is not only characterised by release outside historic distribution (that would be the definition of any introduction), but additionally, and importantly, because of conservation concerns, often caused by climate change. Please refer to IUCN and or Hällfors et al. 2014 who define it even slightly more specifically: <https://journals.plos.org/plosone/article?id=10.1371/journal.pone.0102979>

320-322: please revise this sentence, there is some mistake in the beginning

Caption of Fig. 3: The first sentence/title of the figure is too long.

Decision letter (RSOS-201603.R0)

Dear Mr Morris

On behalf of the Editors, we are pleased to inform you that your Manuscript RSOS-201603 "Roughing it: terrain is crucial in identifying novel translocation sites for the vulnerable brush-tailed rock-wallaby (*Petrogale pencillata*)." has been accepted for publication in Royal Society Open Science subject to minor revision in accordance with the referees' reports. Please find the referees' comments along with any feedback from the Editors below my signature.

Please submit your revised manuscript and required files (see below) no later than 7 days from today's (ie 20-Nov-2020) date. Note: the ScholarOne system will 'lock' if submission of the revision is attempted 7 or more days after the deadline. If you do not think you will be able to meet this deadline please contact the editorial office immediately.

on behalf of Prof Pete Smith (Subject Editor)
openscience@royalsociety.org

Associate Editor Comments to Author:

The referees recommend your paper be modified slightly, and provide helpful suggestions. Please can you amend your work to include these suggestions, and return a revision with a point-by-point response - full details of what you need to include are provided at the end of the decision letter. Congratulations!

Reviewer comments to Author:

Reviewer: 1

Comments to the Author(s)

This is a nicely constructed paper which provides a good model for others wishing to use species distribution models for conservation planning. While the conclusions in this case are unsurprising, they do further refine understanding of the habitat of the Brush-tailed rock-wallaby, and the discussion on potential for introduction to areas outside the current distribution to provide protection from affects of climate change is topical.

I have only a few minor editorial comments and suggestions:

Line 67 - should be wallaby's, not wallabies.

Lines 107-108 – more information is needed on why these two climatic variables [minimum temperature in the coldest month, and annual mean precipitation] were selected from the large pool of potential climate variables.

Line 133 – Sup xxx – to be completed

Line 232 – apostrophe misplaced.

Line 233 – ‘score’ should be plural – there are 2, one for each background.

Line 245 – delete ‘with’

Line 279 – sentence needs recasting as it currently suggests that a dietary assessment from Tasmania exists for the rock-wallaby. The dietary assessment referred to took place at three sites in NSW.

Line 287-288 – need to more directly link your expression ‘the habitat and climate argument’ to the second criterion for assisted migration – the potential to move – assuming that is what is meant.

Line 304 – delete ‘outside their recipient site’, it just repeats the information contributed by ‘areas to which they were introduced’.

Line 314 – insert ‘in’ after ‘increase’.

Line 320 – sentence beginning with ‘It’ needs work – ‘offers’, larger area of potential ?

Line 351 – delete ‘that’ or change ‘as’ to ‘is’.

Line 352 – the terrain in the Grampians is extremely rough and broken. I don’t see that rock-wallabies would be more exposed to predators there than at other occupied sites.

Line 354 – recast to ‘being less accessible to people.’

References - need a careful check, for example the surname Burbidge is missing from number 19

Reviewer: 2

Comments to the Author(s)

The authors describe a study where they used species distribution modeling to define areas to which the brush-tailed rock wallaby could be introduced since its original habitat is being diminished as a consequence of climate change effects.

In general, the manuscript is very well and clearly written, the methods are well justified and described adequately, and the results and conclusions inferred are robust. I see no major concern with the scientific content of the paper.

The study is case-specific and concerns the immediate conservation of a specific species for a specific reason. As such, I feel that the authors could have the potential to reach a more targeted audience through offering the manuscript for publication to a more conservation-oriented paper, like Conservation Biology where it could function as an interesting case-example for similar conservation projects. However, with a bit of effort and considering the open access policy of RSOS, I am sure that the paper has potential to reach readers most interested in this type of case study. The most novel aspect of their paper is the use of terrain characteristics in addition to habitat and climate variables, for modelling the suitable area for the species. As such, it offers a valuable example of how the scientific process can be used for informing practical conservation planning.

I have only a few minor comments, most of which relate to being a bit more specific with the concepts under the rather wide conservation translocation umbrella.

Minor comments:

line 24: could you be more specific on what kind of translocation you refer to? Translocation is an umbrella concept and includes very many different methods. Please refer to the IUCN: <https://www.iucn.org/content/guidelines-reintroductions-and-other-conservation-translocations> A translocation can be unintentional (cf. invasive species) or conducted because of conservation concern (=conservation translocation). I believe you are referring to a conservation introduction, and within that, assisted migration. Throughout the manuscript, please consider where you need to specify these terms more precisely.

line 40. Ditto. This definition is not specific enough, please refer to the IUCN guidelines: "Conservation translocation is the intentional movement and release of a living organism where the primary objective is a conservation benefit."

line 210: visual  visualize?

271-274: Again, please be specific with the definitions. Assisted migration is not only characterised by release outside historic distribution (that would be the definition of any introduction), but additionally, and importantly, because of conservation concerns, often caused by climate change. Please refer to IUCN and or Hällfors et al. 2014 who define it even slightly more specifically: <https://journals.plos.org/plosone/article?id=10.1371/journal.pone.0102979>

320-322: please revise this sentence, there is some mistake in the beginning

Caption of Fig. 3: The first sentence/title of the figure is too long.

===PREPARING YOUR MANUSCRIPT===

- one version identifying all the changes that have been made (for instance, in coloured highlight, in bold text, or tracked changes);
- a 'clean' version of the new manuscript that incorporates the changes made, but does not highlight them. This version will be used for typesetting.

===PREPARING YOUR REVISION IN SCHOLARONE===

Author's Response to Decision Letter for (RSOS-201603.R0)

See Appendix A.

Decision letter (RSOS-201603.R1)

Dear Mr Morris,

It is a pleasure to accept your manuscript entitled "Roughing it: terrain is crucial in identifying novel translocation sites for the vulnerable brush-tailed rock-wallaby (*Petrogale penicillata*)."

 in its current form for publication in Royal Society Open Science.

on behalf of Professor Pete Smith (Subject Editor)
openscience@royalsociety.org

Appendix A

[revised manuscript text omitted]

*Ethics.* No human or animal subjects were included in this study. Therefore, no ethical approvals
were required.

*Data accessibility.* The data and code used to run this experiment are available at
[10.5281/zenodo.4017016](https://doi.org/10.5281/zenodo.4017016)

*Authors' contributions.* S.D.M., C.N.J. and B.W.B conceived of the original idea; S.D.M carried out the
data collection and coding with critical input from C.N.J and B.W.B. All authors contributed to the
writing of this publication and gave final approval for publication.

*Competing interests.* The authors declare no competing interests.

*Funding.* This work was funded by Australian Research Council grants FL160100101 to B. W. B.

*Acknowledgements.* The authors are grateful to all the people who collected and collated the data
used in this research, especially the occurrence data. We acknowledge the traditional owners of the
land on which this data was collection.

**References**

- Caughley, G. Directions in Conservation Biology. 1994:215.
- Lande, R. 1988 Genetics and demography in biological conservation. *Science*. **241**, 1455.
- (doi:10.1126/science.3420403)
- Lande, R., Engen, S., Sæther, B.-E. 1998 Extinction Times in Finite Metapopulation Models with
- Stochastic Local Dynamics. *Oikos*. **83**, 383-389. (doi:10.2307/3546853)
- Shaffer, M. L. 1981 Minimum Population Sizes for Species Conservation. *Bioscience*. **31**, 131-134.
- (doi:10.2307/1308256)
- Mitchell, J. F. B., Lowe, J., Wood, R. A., Vellinga, M. 2006 Extreme events due to human-induced
- climate change. *Phil. Trans. R. Soc. Lond. Ser. A*. **364**, 2117-2133. (doi:10.1098/rsta.2006.1816)
- IUCN/SSC. 2013 Guidelines for Reintroduction and Other Conservation Translocations. Gland,
- Switzerland: IUCN Species Survival Commission.
- Macdonald, D. W., Tattersall, F. H., Rushton, S., South, A. B., Rao, S., Maitland, P., Strachan, R.
- 2000 Reintroducing the beaver (*Castor fiber*) to Scotland: a protocol for identifying and assessing
- suitable release sites. *Anim. Conserv.* **3**, 125-133.
- Lewis, J. C., Hayes, G. E. 2004 Feasibility assessment for reintroducing fishers to Washington:
- Washington Department of Fish and Wildlife, Wildlife Management Program.
- Elith, J., Leathwick, J. R. 2009 Species distribution models: ecological explanation and prediction
- across space and time. *Annual review of ecology, evolution, and systematics*. **40**, 677-697.
- Wilson, C. D., Roberts, D., Reid, N. 2011 Applying species distribution modelling to identify areas
- of high conservation value for endangered species: A case study using *Margaritifera margaritifera*
- (*L.*). *Biol. Conserv.* **144**, 821-829. (doi:10.1016/j.biocon.2010.11.014)
- Bleyhl, B., Sipko, T., Trepel, S., Bragina, E., Leitao, P. J., Radeloff, V. C., Kuemmerle, T. 2015
- Mapping seasonal European bison habitat in the Caucasus Mountains to identify potential
- reintroduction sites. *Biol. Conserv.* **191**, 83-92. (doi:10.1016/j.biocon.2015.06.011)
- Draper, D., Marques, I., Iriondo, J. M. 2019 Species distribution models with field validation, a key
- approach for successful selection of receptor sites in conservation translocations. *Glob. Ecol.*
- *Conserv.* **19**, e00653. (doi:10.1016/j.gecco.2019.e00653)
- Zurell, D., Thuiller, W., Pagel, J., Cabral, J. S., Münkemüller, T., Gravel, D., Dullinger, S., Normand,
- S., Schiffers, K. H., Moore, K. A., *et al.* 2016 Benchmarking novel approaches for modelling
- species range dynamics. *Global Change Biol.* **22**, 2651-2664. (doi:10.1111/gcb.13251)
- Briscoe, N. J., Elith, J., Salguero-Gómez, R., Lahoz-Monfort, J. J., Camac, J. S., Giljohann, K. M.,
- Holden, M. H., Hradsky, B. A., Kearney, M. R., McMahon, S. M., *et al.* 2019 Forecasting species range
- dynamics with process-explicit models: matching methods to applications. *Ecol. Lett.* **22**, 1940-1956.
- (doi:10.1111/ele.13348)
- Buckley, L. B., Urban, M. C., Angilletta, M. J., Crozier, L. G., Rissler, L. J., Sears, M. W. 2010 Can
- mechanism inform species' distribution models? *Ecol. Lett.* **13**, 1041-1054. (doi:10.1111/j.1461-
- 0248.2010.01479.x)
- Briscoe, N. J., Kearney, M. R., Taylor, C. A., Wintle, B. A. 2016 Unpacking the mechanisms
- captured by a correlative species distribution model to improve predictions of climate refugia.
- *Global Change Biol.* **22**, 2425-2439. (doi:10.1111/gcb.13280)
- Guisan, A., Tingley, R., Baumgartner, J. B., Naujokaitis-Lewis, I., Sutcliffe, P. R., Tulloch, A. I.,
- Regan, T. J., Brotons, L., McDonald-Madden, E., Mantyka-Pringle, C., *et al.* 2013 Predicting species
- distributions for conservation decisions. *Ecol. Lett.* **16**, 1424-1435. (doi:10.1111/ele.12189)
- Hallfors, M. H., Aikio, S., Fronzek, S., Hellmann, J. J., Rytteri, T., Heikkinen, R. K. 2016 Assessing
- the need and potential of assisted migration using species distribution models. *Biol. Conserv.* **196**,
- 60-68. (doi:10.1016/j.biocon.2016.01.031)

Woinarski, J., Burbidge, A. A. *Petrogale penicillata*. The IUCN Red List of Threatened Species
2016: e.T16746A21955754. . 2016 [cited 2020 15 June]; Available from:
<https://dx.doi.org/10.2305/IUCN.UK.2016-1.RLTS.T16746A21955754.en>
Ward, M., Tulloch, A. I. T., Radford, J. Q., Williams, B. A., Reside, A. E., Macdonald, S. L., Mayfield,
H. J., Maron, M., Possingham, H. P., Vine, S. J., *et al.* 2020 Impact of 2019–2020 mega-fires on
Australian fauna habitat. *Nat. Ecol. Evol.* **4**, 1321-1326. (doi:10.1038/s41559-020-1251-1)
Jarman, P. J., Bayne, P. 1997 Behavioural ecology of *Petrogale penicillata* in relation to
conservation. *Aust. Mammal.* **19**, 219-228.
Short, J. 1982 Habitat Requirements of the Brush-Tailed Rock-Wallaby, *Petrogale penicillata*, in
New South Wales. *Wildl. Res.* **9**, 239-246. (doi:10.1071/WR9820239)
Short, J., Milkovits, G. 1990 Distribution and Status of the Brush-Tailed Rock-Wallaby in South-
Eastern Australia. *Wildl. Res.* **17**, 169-179.
Murray, J. V., Low Choy, S., McAlpine, C. A., Possingham, H. P., Goldizen, A. W. 2008 The
importance of ecological scale for wildlife conservation in naturally fragmented environments: A
case study of the brush-tailed rock-wallaby (*Petrogale penicillata*). *Biol. Conserv.* **141**, 7-22.
(doi:10.1016/j.biocon.2007.07.020)
Menkhorst, P., Hynes, E. 2011 National recovery plan for the Brush-tailed Rock-wallaby *Petrogale*
*penicillata*
Woinarski, J. C., Burbidge, A., Harrison, P. 2014 Brush-tailed Rock-wallaby. In *The action plan for*
*Australian mammals 2012*. (J. C. Woinarski), pp. 422-425. Collingwood, Australia: CSIRO publishing.
Eldridge, M., Close, R. 2008 Brush-tailed rock-wallaby *Petrogale penicillata*. *The mammals of*
*Australia*. 382-384.
Taggart, D. A., Schultz, D. J., Corrigan, T. C., Schultz, T. J., Stevens, M., Panther, D., White, C. R.
2016 Reintroduction methods and a review of mortality in the brush-tailed rock-wallaby, Grampians
National Park, Australia. *Aust. J. Zool.* **63**, 383-397. (doi:10.1071/ZO15029)
Warburton, B. W. S., R.M.F. 1995 Brush-tailed Rock-wallaby. In *The Handbook of New Zealand*
*Mammals*. (C. King), pp. 58-64. Auckland: Oxford University Press.
Lazell, J. D., Sutterfield, T. W., Giezantner, W. D. 1984 The population of rock wallabies (genus
*Petrogale*) on Oahu, Hawaii. *Biol. Conserv.* **30**, 99-108. (doi:10.1016/0006-3207(84)90060-0)
Eldridge, M. D. B., Neaves, L. E., Faris, J., Soderquist, T. 2018 Genetic affinities of a remnant
population of the brush-tailed rock-wallaby (*Petrogale penicillata*) in Mt Kaputar National Park,
northern New South Wales. *Aust. Mammal.* **40**, 112-117. (doi:10.1071/AM16051)
Murray, J. V., Low Choy, S., McAlpine, C. A., Possingham, H. P., Goldzien, A. W. 2011 Evaluating
model transferability for a threatened species to adjacent areas: Implications for rock-wallaby
conservation. *Austral Ecol.* **36**, 76-89. (doi:10.1111/j.1442-9993.2010.02122.x)
Ashcroft, M. B., Cavanagh, M., Eldridge, M. D. B., Gollan, J. R. 2014 Testing the ability of
topoclimatic grids of extreme temperatures to explain the distribution of the endangered brush-
tailed rock-wallaby (*Petrogale penicillata*). *J. Biogeogr.* **41**, 1402-1413. (doi:10.1111/jbi.12298)
Phillips, S. J., Anderson, R. P., Schapire, R. E. 2006 Maximum entropy modeling of species
geographic distributions. *Ecol. Model.* **190**, 231-259.
Elith, J., H. Graham, C., P. Anderson, R., Dudík, M., Ferrier, S., Guisan, A., J. Hijmans, R.,
Huettmann, F., R. Leathwick, J., Lehmann, A., *et al.* 2006 Novel methods improve prediction of
species' distributions from occurrence data. *Ecography.* **29**, 129-151. (doi:10.1111/j.2006.0906-
7590.04596.x)
Elith, J., Phillips, S. J., Hastie, T., Dudík, M., Chee, Y. E., Yates, C. J. 2011 A statistical explanation of
MaxEnt for ecologists. *Divers. Distrib.* **17**, 43-57. (doi:10.1111/j.1472-4642.2010.00725.x)
Merow, C., Smith, M. J., Silander, J. A. 2013 A practical guide to MaxEnt for modeling species'
distributions: what it does, and why inputs and settings matter. *Ecography.* **36**, 1058-1069.
Department of the Environment. Interim Biogeographic Regionalisation for Australia v.7 (IBRA).
2012.

Barbet-Massin, M., Jiguet, F., Albert, C. H., Thuiller, W. 2012 Selecting pseudo-absences for
species distribution models: how, where and how many? *Methods Ecol. Evol.* **3**, 327-338.
(doi:10.1111/j.2041-210X.2011.00172.x)

Beaumont, L. J., Hughes, L., Poulsen, M. 2005 Predicting species distributions: use of climatic
parameters in BIOCLIM and its impact on predictions of species' current and future distributions.
*Ecol. Model.* **186**, 251-270. (doi:10.1016/j.ecolmodel.2005.01.030)

Hijmans, R. J. 2019 Introduction to the 'raster' package (version 3.0-7).

Department of Environment. Australia—Present Major Vegetation Groups—NVIS Version 5.1
(Albers 100 m analysis product). 2012.

Grinnell, J. 1917 Field tests of theories concerning distributional control. *Am. Nat.* **51**, 115-128.

Andrewartha, H. G., Birch, L. C. 1954 *The distribution and abundance of animals*. University of
Chicago press.

MacArthur, R. H. 1984 *Geographical ecology: patterns in the distribution of species*. Princeton
University Press.

Hutchinson, M., Kesteven, J., Xu, T. Monthly total precipitation: ANUClimate 1.0, 0.01 degree,
Australian Coverage, 1970–2012. In: C. Australian National University, Australia. Obtained from
<http://dap.nci.org.au>, made available by the Ecosystem Modelling and Scaling Infrastructure
(eMAST, <http://www.emast.org.au>) of the Terrestrial Ecosystem Research Network (TERN,
<http://www.tern.org.au>), ed. 2014.

Dormann, C. F., Elith, J., Bacher, S., Buchmann, C., Carl, G., Carré, G., Marquéz, J. R. G., Gruber, B.,
Lafourcade, B., Leitão, P. J., *et al.* 2013 Collinearity: a review of methods to deal with it and a
simulation study evaluating their performance. *Ecography*. **36**, 27-46. (doi:10.1111/j.1600-
0587.2012.07348.x)

Elith, J., Kearney, M., Phillips, S. 2010 The art of modelling range-shifting species. *Methods Ecol.*
*Evol.* **1**, 330-342. (doi:10.1111/j.2041-210X.2010.00036.x)

Phillips, S. J., Dudík, M. 2008 Modeling of species distributions with Maxent: new extensions and
a comprehensive evaluation. *Ecography*. **31**, 161-175. (doi:10.1111/j.0906-7590.2008.5203.x)

Bluff, L. A., Clausen, L., Hill, A., Bramwell, M. D. 2011 A decade of monitoring the remnant
Victorian population of the brush-tailed rock-wallaby (*Petrogale penicillata*). *Aust. Mammal.* **33**, 195-
201. (doi:10.1071/AM10037)

Allouche, O., Tsoar, A., Kadmon, R. 2006 Assessing the accuracy of species distribution models:
prevalence, kappa and the true skill statistic (TSS). *J. Appl. Ecol.* **43**, 1223-1232. (doi:10.1111/j.1365-
2664.2006.01214.x)

Naimi, B., Araújo, M. B. 2016 sdm: a reproducible and extensible R platform for species
distribution modelling. *Ecography*. **39**, 368-375. (doi:10.1111/ecog.01881)

Baumgartner, J., Wilson, P., Esperón-Rodríguez, M. 2017 rmaxent: Tools for working with Maxent
in R. *R package version 0.8*.

Forsyth, D. M., Woodford, L., Moloney, P. D., Hampton, J. O., Woolnough, A. P., Tucker, M. 2014
How does a carnivore guild utilise a substantial but unpredictable anthropogenic food source?
Scavenging on hunter-shot ungulate carcasses by wild dogs/dingoes, red foxes and feral cats in
south-eastern Australia revealed by camera traps. *PLoS one*. **9**, e97937-e97937.
(doi:10.1371/journal.pone.0097937)

Eldridge, M., Close, R. 1992 Taxonomy of rock wallabies, *Petrogale* (*Marsupialia*, *Macropodidae*).
1. A revision of the Eastern *Petrogale* with the description of 3 new species. *Aust. J. Zool.* **40**, 605-
625.

Peters, K. J., Saltre, F., Friedrich, T., Jacobs, Z., Wood, R., McDowell, M., Ulm, Bradshaw, C. J. A.
2019 *FosSahul 2.0 database and R code*.

Seddon, P. J., Armstrong, D. P., Soorae, P., Launay, F., Walker, S., Ruiz-Miranda, C. R., Molur, S.,
Koldewey, H., Kleiman, D. G. 2009 The Risks of Assisted Colonization. *Conserv. Biol.* **23**, 788-789.
(doi:10.1111/j.1523-1739.2009.01200.x)

- Jarman, P., Phillips, C. 1989 Diets in a community of macropod species. *Kangaroos, wallabies and*
*rat-kangaroos*. **1**, 143-149.
- Tuft, K. D., Crowther, M. S., McArthur, C. 2011 Multiple scales of diet selection by brush-tailed
rock-wallabies (*Petrogale penicillata*). *Aust. Mammal.* **33**, 169-180. (doi:10.1071/AM10041)
- Hallfors, M. H., Aikio, S., Schulman, L. E. 2017 Quantifying the need and potential of assisted
migration. *Biol. Conserv.* **205**, 34-41. (doi:10.1016/j.biocon.2016.11.023)
- Mowbray, S. C. 2002 Eradication of introduced Australian marsupials (brushtail possum and
brushtailed rock wallaby) from Rangitoto and Motutapu Islands, New Zealand. In *Turning the tide:*
*the eradication of invasive species*. (C. R. Veitch, M. N. Clout), pp. 226-232. IUCN, Gland, Switzerland
and Cambridge, UK.: IUCN SSC Invasive Species Specialist Group.
- Hawaii New Now. Animals Go Wild! The wallabies of Kalihi Valley. 2009 [cited 2020
15/06/2020]; Available from: [https://www.hawaiinewsnow.com/story/10366453/animals-go-wild-](https://www.hawaiinewsnow.com/story/10366453/animals-go-wild-the-wallabies-of-kalihi-valley/)
[the-wallabies-of-kalihi-valley/](https://www.hawaiinewsnow.com/story/10366453/animals-go-wild-the-wallabies-of-kalihi-valley/)
- Rose, R. K., Pemberton, D. A., Mooney, N. J., Jones, M. E. 2017 *Sarcophilus harrisii*
(Dasyuromorphia: Dasyuridae). *Mammalian Species*. **49**, 1-17. (doi:10.1093/mspecies/sex001)
- Kinneer, J. E., Onus, M. L., Bromilow, R. N. 1988 Fox control and rock-wallaby population
dynamics. *Wildl. Res.* **15**, 435-450. (doi:10.1071/WR9880435)
- Kinneer, J. E., Onus, M. L., Sumner, N. R. 1998 Fox control and rock-wallaby population dynamics;
II. An update. *Wildl. Res.* **25**, 81-88. (doi:10.1071/WR96072)
- Grose, M., Barnes-Keoghan, I., Corney, S., White, C., Holz, G., Bennett, J., Gaynor, S., Bindoff, N.
2010 Climate futures for Tasmania: general climate impacts technical report.
- Thalmann, S., Peck, S., Wise, P., Potts, J. M., Clarke, J., Richley, J. 2016 Translocation of a top-
order carnivore: tracking the initial survival, spatial movement, home-range establishment and
habitat use of Tasmanian devils on Maria Island. *Aust. Mammal.* **38**, 68-79. (doi:10.1071/am15009)
- Tasmania Parks and Wildlife Service. Bishop and Clerk. 2020 [cited 2020 15/06/2020]; Available
from: <https://parks.tas.gov.au/explore-our-parks/maria-island-national-park/bishop-and-clerk>
- White, L. C., Saltré, F., Bradshaw, C. J. A., Austin, J. J. 2018 High-quality fossil dates support a
synchronous, Late Holocene extinction of devils and thylacines in mainland Australia. *Biol. Lett.* **14**,
20170642. (doi:10.1098/rsbl.2017.0642)
- Cheng, J., Karambelkar, B., Xie, Y. 2018 leaflet: Create Interactive Web Maps with the JavaScript
'Leaflet' Library: R package version 2.0.2.2018.
- Caughley, J., Bayliss, P., Giles, J. 1984 Trends in Kangaroo Numbers in Western New South Wales
and their relation to Rainfall. *Wildl. Res.* **11**, 415-422. (doi:10.1071/WR9840415)
- Banks, P. B., Newsome, A. E., Dickman, C. R. 2000 Predation by red foxes limits recruitment in
populations of eastern grey kangaroos. *Austral Ecol.* **25**, 283-291. (doi:10.1046/j.1442-
9993.2000.01039.x)
- Sharp, A., McCallum, H. 2010 The decline of a large yellow-footed rock-wallaby (*Petrogale*
*xanthopus*) colony following a pulse of resource abundance. *Aust. Mammal.* **32**, 99-107.
(doi:10.1071/AM08113)
- Toor, M. L. v., Arriero, E., Holland, R. A., Huttunen, M. J., Juvaste, R., Müller, I., Thorup, K.,
Wikelski, M., Safi, K. 2017 Flexibility of habitat use in novel environments: insights from a
translocation experiment with lesser black-backed gulls. *R. Soc. Open Sci.* **4**, 160164.
(doi:10.1098/rsos.160164)
- Richardson, D. M., Hellmann, J. J., McLachlan, J. S., Sax, D. F., Schwartz, M. W., Gonzalez, P.,
Brennan, E. J., Camacho, A., Root, T. L., Sala, O. E., *et al.* 2009 Multidimensional evaluation of
managed relocation. *Proc. Natl Acad. Sci.* **106**, 9721-9724. (doi:10.1073/pnas.0902327106)
